# Improved exclusive breastfeeding rates in preterm infants after a neonatal nurse training program focusing on six breastfeeding-supportive clinical practices

Ragnhild Maastrup[1,2]*, Ane L. Rom[2], Sisse Walloee[3], Helle B. Sandfeld[4], Hanne Kronborg[5]

**1** Department of Neonatology, Knowledge Centre for Breastfeeding Infants with Special Needs, Rigshospitalet, Copenhagen University Hospital, Copenhagen, Denmark, **2** Research Unit Women's and Children's Health, Juliane Marie Centre, Rigshospitalet, Copenhagen University Hospital, Copenhagen, Denmark, **3** Dept of Clinical Research, OPEN—Patient data Explorative Network, University of Southern Denmark, Odense, Denmark, **4** Department of Paediatrics, Randers Hospital, Randers, Denmark, **5** Department of Public Health, Section for Nursing, Aarhus University, Aarhus, Denmark

* ragnhild.maastrup@regionh.dk

**Data Availability Statement:** The authors confirm that, the data are available on figshare DOI 10.6084/m9.figshare.13251347, https://figshare.com/

## Abstract

### Background

Early breast milk expression, prolonged skin-to-skin contact, rooming-in, use of test-weighing and minimizing use of pacifiers are positively associated with exclusive breastfeeding of preterm infants, whereas use of nipple shields is negatively associated.

### Aim

To test whether a training program for neonatal nurses with a focus on these six breastfeeding-supportive clinical practices affects the rate of preterm infants exclusively breastfed at discharge to home, the postmenstrual age at establishment of exclusive breastfeeding, and maternal self-reported use of the practice in the neonatal intensive care unit, the.

### Methods

A quasi-experimental multi-centre intervention study from 2016–2019 including a control group of 420 preterm mother-infant dyads, an intervention with a training program for neonatal nurses and implementation of weekly breastfeeding meetings for neonatal nurses, and an intervention group of 494 preterm mother-infant dyads.

### Results

Significantly more preterm infants in the intervention group were exclusively breastfed at discharge to home (66.6%) than in the control group (58.1%) p = 0.008. There was no significant difference in postmenstrual age at establishment of exclusive breastfeeding between control and intervention group (37.5 vs.37.8 weeks, p = 0.073). Compared to the control group the number of infants continuing daily skin-to-skin contact after incubator care

articles/dataset/Data_Improvement_of_exclusive_
breastfeeding_in_preterm_infants_2/13251347.

**Funding:** The authors disclosed receipt of the
following financial support for the research and
authorship of this article: Funding to support the
research was received from Novo Nordisk
Foundation, grant number NNF15OC0018156
https://novonordiskfonden.dk/en/ (RM), Lundbeck
Foundation, grant number F-23137-01 https://
www.lundbeckfonden.com/ (RM), Helsefonden,
grant number 16-B-0059 https://helsefonden.dk/
(RM), Department of Neonatology, Rigshospitalet,
Copenhagen University Hospital https://www.
rigshospitalet.dk/afdelinger-og-klinikker/
julianemarie/neonatalklinikken/Sider/default.aspx
(RM), and Rigshospitalet, Copenhagen University
Hospital, Research Committee https://www.
rigshospitalet.dk/ (RM). The funders had no role in
study design, data collection and analysis, decision
to publish, or preparation of the manuscript.

**Competing interests:** I have received funding from
Novo Nordisk Foundation and Lundbeck
Foundation. This does not alter our adherence to
PLOS ONE policies on sharing data and materials.

increased (83.2% vs. 88.3%, p = 0.035), infants using a nipple shield decreased (61.8% vs. 54.2%, p = 0.029), and the number of mothers initiating breast milk expression before six hours post-partum increased (32.6% vs. 42.4%, p = 0.007). There was a significant correlation between percentage of neonatal nurses participating in the breastfeeding training program and changes in exclusive breastfeeding rates (Pearson Correlation 0.638, p = 0.047).

## Conclusion

Exclusive breastfeeding rates in preterm infants and maternal self-reported use of breastfeeding-supportive practices increased by training neonatal nurses in the six clinical practices. It is important to include all nurses in the breastfeeding training program to ensure positive effect on exclusive breastfeeding rates.

## Introduction/Background

Preterm infants often suffer from respiratory, gastrointestinal, immunological, and neurological morbidity [1]. In general, breastfeeding provides health benefits for all infants [2], and specifically protects preterm infants from severe illness such as sepsis, necrotizing enterocolitis, and retinopathy of prematurity [3–5] and improve neurodevelopment [6]. Despite this, studies have shown that fewer preterm infants than term infants are breastfed exclusively at discharge; 45% of preterm infants in Sweden [7], 68% in Denmark [8] compared to 75% of 10-days-old term infants [9]. Countries outside of Scandinavia often have even lower exclusive breastfeeding rates at discharge; 55% of preterm infants in an Australian study [10], 23% in a Japanese study [11], and for very preterm infants 22% in an US study [12], and 20% in a European study of 11 countries [13].

Establishing exclusive direct breastfeeding is crucial for preterm infants. It is associated with longer duration of breastmilk feeding as compared to the duration of breastmilk feeding for infants feeding some or all of their exclusive breastmilk diet from a bottle at the time of discharge [14,15].

In a cohort study including 1488 preterm infants from all neonatal intensive care units (NICUs) in Denmark in 2009–11, we previously found early breast milk expression, use of test-weighing, and minimizing use of pacifiers positively associated with exclusive breastfeeding at discharge; whereas use of nipple shields was negatively associated [8]. Regarding earlier establishment of exclusive breastfeeding we found a positive association with early breast milk expression, minimizing use of pacifiers, prolonged skin-to-skin contact, and rooming-in [14]. In other studies, these six clinical practices (i.e. early breast milk expression, use of test-weighing, minimizing use of pacifiers, use of nipple shields, prolonged skin-to-skin contact, and rooming-in) have also been found to play a role: Sufficient breast milk supply has shown to be essential in establishing exclusive breastfeeding [16], and early initiation of breast milk expression resulted in higher amounts of breast milk [17–19] and better possibilities of establishing exclusive breastfeeding [8]. Test-weighing has been associated with successful establishment of exclusive breastfeeding in two studies [8,20]. Findings from studies of pacifier use are contradictive with some suggesting that pacifier use in preterm infants is associated with earlier transition to exclusive breastfeeding [21] and that minimizing the use of pacifiers during the period of breastfeeding establishment is protective to exclusive breastfeeding [8,14]. The use of nipple shields in preterm infants has been associated with a more than doubled risk of not

establishing exclusive breastfeeding [8,22], and with increased risk of earlier cessation of exclusive breastfeeding [22,23]. Rooming-in of mothers with their preterm infant has been related to earlier establishment of exclusive breastfeeding, more breastfeeding at discharge, and breastfeeding for a longer period [14,24,25]. Finally, preterm infants who had more skin-to-skin contact established exclusive breastfeeding earlier, suggesting a dose-response effect [14,26,27]. Improvement of these six clinical practices is needed to protect breastfeeding in preterm infants and are supported by The Baby-friendly Hospital Initiative for Neonatal wards [28–30].

Previously, it has been shown, that training of health care staff can change clinical practice and guidance in neonatal wards and thereby change practice of mothers of preterm infants [17,31]. Therefore, the aim of this study was to test whether a training program for neonatal nurses with focus on the six breastfeeding-supportive clinical practices affects the rate of preterm infants exclusively breastfed at discharge from the NICU, the postmenstrual age at establishment of exclusive breastfeeding and maternal self-reported use of the practice in the NICU. The aim was achieved.

## Materials and methods

### Ethic statement

Nurse managers at the NICUs and the mothers of the preterm infants were informed that participation was voluntary. A written consent was signed by nurse managers from each participating NICU and another by each participating mother. The study was conducted in accordance with the Declaration of Helsinki [32] and approved by The Danish Data Protection Agency (Journal number2012-58-0004). It was pointed out to the participating neonatal nurses that breastfeeding support should focus on supporting mothers in fulfilling their own goals for breastfeeding, since it will always be the mother's individual decision whether she will breastfeed and for how long.

### Definition of terminology

Exclusive breastfeeding was defined as an infant solely feeding directly at and from the breast [33] on the day of discharge and could include medication and vitamins. This definition does not count for what or how the infant has been fed before exclusive breastfeeding was established. Partial breastfeeding included other feeding methods in addition to breastfeeding (i.e. bottle, cup, lact-aid) regardless of content. Breastmilk feeding included breastfeeding and breastmilk given by other feeding methods.

Discharge was defined as the end of the total period of care from the NICU. If the infant was in an Early Discharge Program, the discharge date was defined as the day the Early Discharge Program ended.

Postmenstrual age is gestational age (GA) plus time elapsed from birth.

All participating neonatal and paediatric wards provided respiratory support (nasal CPAP or mechanical ventilation) for new-born infants and are for practical reasons referred to as NICUs.

### Design and settings

A multi-centre study using a quasi-experimental intervention design was conducted. From 2016, three phases were performed: 1) a pre-intervention phase with participation of preterm mother-infant dyads born from October 2016 to July 2017 serving as a control group, 2) an intervention phase including a training program for neonatal nurses, implementation of

weekly breastfeeding meetings for neonatal nurses, and optimizing their clinical practices, and 3) a post-intervention phase with participation of preterm mother-infant dyads born from February to December 2018 serving as an intervention group.

In Denmark, health care is free of charge. Parents in Denmark have the right to paid parental leave for a total of 52 weeks, 18 weeks solely for the mother and 2 weeks solely for the partner. The reminding weeks can freely be shared between the parents [34]. About 97% of Danish infants initiate breastfeeding [8] and 61% are exclusive breastfed for four months [35]. Preterm infants less than 35 gestational weeks are always admitted to a NICU, whereas infants born ≥35 gestational weeks are only admitted to a NICU if they need additional treatment and care. Most of the Danish NICUs have an early discharge program with transition from tube-feeding to breastfeeding at home and with contact to the NICU twice a week. In general, preterm infants are hospitalized or in early discharge program until exclusive breastfeeding is established or given up and mixed or bottle-feeding established. Guidance of mothers in establishing breastfeeding is a core element of neonatal nursing in Denmark.

## Participants

All 17 Danish NICUs and one children's department, which as a routine cared for preterm infants during breastfeeding establishment, were invited. Ten NICUs participated in the whole study, among them all four high intensive NICUs (level 3B and 3C). Three NICUs refused to participate due to other ongoing intervention studies with breastfeeding as outcome, two NICUs refused because they had other priorities. One NICU participated only in the pre-intervention, as they could not allocate resources for the training program and the post-intervention, two NICUs participated only in the intervention phase and post-intervention phase as they had to complete other ongoing studies first. The 13 participating NICUs had a mean of 18.7 beds and 54 neonatal nurses (range 24–140), all NICUs had access to donor human milk, all but one had an early discharge program. No hospitals held a valid BFHI designation as the program was closed in Denmark in 2008. Rooming-in was possible in all NICUs, in eight the mother could room-in for the infant's whole NICU stay, in three NICUs at least 50% of the infant's NICU stay and in two NICUs at least the last night before infant discharge.

Inclusion criteria for participating in the control and intervention groups were preterm infants (<37 gestational weeks) admitted to a NICU within five days of birth. Exclusion criteria were infant discharge from NICU to maternity or paediatric wards other than neonatal, mothers of preterm infants not capable of reading the questionnaires in Danish with or without help from the family, or history of drug abuse, which precluded the recommendation to breastfeed. Mothers of preterm infants who were particularly vulnerable e.g. giving infant away for foster care, psychiatric problems, or too stressed by the infant's severe conditions were not invited to participate in the study. In the control group, infants could be transferred to non-participating NICUs and the mothers were still sent a questionnaire at the time of infant's discharge to home. In the intervention group, infants had to be admitted to a participating NICU within the first days and discharged to home from a participating NICU.

## Intervention

**Training program for neonatal nurses.** After the pre-intervention phase, the neonatal nurses received one day of training (seven hours) in evidence-based breastfeeding support about the six clinical practices associated with exclusive breastfeeding of preterm infants (see S1 File). Each NICU had from one to six days of training, depending on the size of the NICU, in order to cover training of all neonatal nurses. The dates were planned in collaboration with the NICU nurse managers, who scheduled the nurses for the training. The training was

provided by the principal investigator in all NICUs. Through training the nurses should be able to:

- Ensure early information and support for mothers before or immediately after premature delivery, so that breast milk expression could begin before six hours after delivery.

- Work towards more co-admissions of mothers into the NICU i.e. earlier rooming-in.

- Minimize the use of pacifiers during breastfeeding establishment and limit the use of nipple shields in her own practice and in her guidance of mothers.

- Support test-weighing according to the mothers' wishes.

- Facilitate and encourage daily skin-to-skin contact between preterm infants and parents throughout the hospitalization to the extent that the infant is sufficiently stable, and the parents wish to participate.

- Prioritize extra attention to groups at particular risk of not breastfeeding exclusively e.g. infants born before 32 gestational weeks, multiples, mothers with lower education, shorter breastfeeding experiences and smokers.

Each NICU were presented to their own data on the six clinical practices from the control group and from a former study (2009–2011) [8,14] and could benchmark their results with the national results. The training sessions included brainstorms of how to facilitate implementation of the six practices locally and resulted in local and national idea catalogues. Two examples of ideas from the brainstorms were 1) having the breast pump kit ready beside the incubator for new infants, and 2) keeping the nipple shields in a less accessible location (i.e. behind a locked door, on the back shelf) with a sign in front of them saying "Did you try alternatives to solve the breastfeeding problem?" and a guideline of alternatives.

**Breastfeeding meetings.** In addition to training of neonatal nurses, the NICUs were encouraged to initiate weekly breastfeeding meetings for neonatal nurses led by an International Board Certified Lactation Consultant, if available, or a neonatal nurse with special responsibility for breastfeeding in the NICU. The purpose of the breastfeeding meetings was to ensure that the neonatal nurses inspired one another and maintained focus on adjustments of their clinical practice. It included a summary of all mother-infant dyads in relation to breastfeeding such as early breastfeeding initiation/attempts, breast milk expression (initiation and supply), skin-to-skin contact (early, prolonged and continuous), transition from tube-feeding to breastfeeding, rooming-in, and use of pacifiers, nipple shields and test-weighing.

Finally, a small poster was distributed to the participating NICUs with short statements intending to support breastfeeding (see S2 File). All NICUs displayed the poster where parents could see it, some NICUs also gave the poster to all parents.

## Data collection

In the pre- and post-intervention phases two questionnaires adapted from our previous study of breastfeeding preterm infants were used with few revisions and pilot tested by one mother [7]. Content in first questionnaire: Demographic data on infant and mother (infant GA, birth weight, maternal age, education), experiences with and plans for breastfeeding, first breastfeeding attempt, breast milk expression, and skin-to-skin contact. Content in second questionnaire: Timing of and method at full oral feeding, feeding at discharge (exclusive breastfeeding, partial breastfeeding, or no breastfeeding), breastfeeding problems, reasons for not establishing exclusive breastfeeding, use of nipple shield, pacifier, and test-weighing, continued skin-

to-skin contact and breast milk pumping, rooming-in, early discharge programs, breastfeeding support, and infant respiratory support (see questionnaires in full in S3 and S4 Files).

The questionnaires were available in an online format that automatically excluded or included more questions depending on the mothers' answers. The first questionnaire was sent to the mother approximately one week after delivery, the second at the infant's discharge to home. Mothers were reminded to answer the questionnaire alternately by text message and e-mail at least three times. To improve participation rate in the intervention group, most of the mothers who did not answer the questionnaire within three reminders were called by phone to complete the questionnaire as a telephone interview. The two questionnaires included a maximum of 43 and 62 questions, respectively.

To monitor the process of the intervention, the number of nurses participating in the training was registered for each NICU together with the total number of nurses working in the NICU. In each NICU a chart was filled out by the person leading the breastfeeding meeting and included the date, topics discussed, number of participants, and length in minutes. The concept was pilot tested in one NICU.

Two or more contact nurses were appointed in each participating NICU. They registered all preterm infants admitted to the NICU during the study periods, informed and included mother-infant dyads in the study, and performed and registered the breastfeeding meetings for neonatal nurses. Encouraging and supportive e-mails were sent to the contact nurses and nurse managers at two to three months intervals to improve adherence to the study. The contact nurses could contact the principal investigator with questions and need for support.

### Process and outcome variables

The process variables were the proportion of neonatal nurses that have gone through the training program in each ward and registration of the breastfeeding meetings (frequency, length, number of participants, and topics).

The primary outcome was exclusive breastfeeding rates at discharge from the NICU to home and changes between control and intervention groups. Secondary outcomes were changes in postmenstrual age at establishment of breastfeeding and changes in the frequency of maternal self-reported use of the six clinical practices: nipple shield use, test-weighing, minimizing the use of pacifier, daily skin-to-skin contact after incubator care, early breast milk expression, and mother-infant rooming-in (staying overnight in same room) for the whole NICU stay.

### Statistics

To detect an expected difference in exclusive breastfeeding rates from 68% to 76%, 500 infants should be included in the control group and 500 in the intervention group ($p = 0.05$, power 80%). Rates from a former study of exclusive breastfeeding in Danish NICUs were used [7]. Although the inclusion period for the control group was extended due to lost cases, the number of infants with information on exclusive breastfeeding was 421. Based on the exclusive breastfeeding rate in the control group (58% of 421 infants), a new power calculation was performed estimating that with a clinically significant increase of 9 percentage points the intervention group should include 488 infants ($p = 0.05$, power 80%).

Descriptive statistics (numbers and percentages) were used to describe characteristics of infants and mothers in the control and the intervention group, respectively. To detect differences in characteristics, primary, and secondary outcomes between groups, Pearson's Chi-Square test for categorical variables and One-way ANOVA for continuous variables were applied. All infants in the control group were included in the analyses irrespective of the ward's participation in the intervention, and all infants in the intervention group were

included in the analyses irrespective of the wards' implementation of breastfeeding meetings. Postmenstrual age at establishment of exclusive breastfeeding was calculated from the date the infant did not get any feeds with any methods other than breastfeeding.For the outcomes exclusive breastfeeding at discharge from the NICU and postmenstrual age at establishment of exclusive breastfeeding, comparisons were made between infants in the intervention and in the control group. Similarly, for the clinical practices; use of a nipple shield, minimizing use of pacifier during breastfeeding establishment, and daily skin-to-skin contact after incubator care, comparisons were made between infants, whereas when assessing changes in first breast milk expression < 6 hours after birth, rooming in of mothers and using test-weighing at most breastfeeds, comparisons were made between mothers. To test correlation between the proportions of neonatal nurses participating in the training in each NICU and the changes in breastfeeding rates from control to intervention group, a Scatterplot was performed, and Pearson's correlation coefficient was calculated. The same approach was used to test correlation between changes in maternal self-reported use of clinical practices and how frequent the practices were discussed at breastfeeding meetings.

A number of sensitivity analyses were performed. To account for a potential dependency between observations all analyses were repeated with only one infant per mother. Furthermore, the analyses assessing clinical practice related to the mother were repeated with the number of infants instead of mothers. To ensure comparable values between NICUs of different sizes and different ways of performing breastfeeding meetings, number of participants were added for all breastfeeding meetings in each NICU and divided with number of neonatal nurses working in the NICU resulting in an attendance nurse ratio. Total duration of breastfeeding meetings was calculated per nurse working in each NICU.

A post-hoc analysis was performed after a significant difference was found between numbers of extremely preterm (<28 weeks) infants in the control and in the intervention group; to test the Linear-by-Linear Association in the exclusive breastfeeding rates among GA groups in the intervention and the control group exclusive breastfeeding rates at discharge were analysed between gestational age groups using the Mantel-Haenszel test of trend.

SPSS version 25 was used for relevant statistical analyses. Values of p <0.05 were considered statistically significant.

## Results

Of the 786 infants eligible for the control group, consent was obtained for 72% (569) of the infants, and information on breastfeeding at discharge was obtained for 53% (420 infants of 369 mothers). Of the 1033 infants eligible for the intervention group, consent was obtained for 65% (674) of the infants and breastfeeding at discharge was obtained for 48% (494 infants of 423 mothers, Fig 1). Of those who consented, breastfeeding at discharge was obtained for 74% and 73%, respectively.

The control group and intervention group were comparable except for a significant difference in GA groups with fewer extremely preterm infants in the intervention group (p = 0.027, Table 1), GA ranged $24^3$–$36^6$ in control group and $23^2$–$36^6$ in intervention group. More than 25% of the infants in both groups were multiples. Almost all mothers planned to breastfeed, more than 60% in both groups were first-time mothers, and nearly half of the mothers were older than 30 years (Table 1). Most mothers perceived the help, support, and encouragement from the nursing staff as sufficient (86% in control group and 90% in intervention group).

Significantly more preterm infants in the intervention group were exclusively breastfed at discharge to home than in the control group (58.1% in control vs. 66.6% in intervention) p = 0.008 (Table 2). There was no significant difference in postmenstrual age at establishment

Flowchart

**Fig 1. Flowchart.** M = mothers, PI = preterm infants, Q1 = questionnaire 1, Q2 = questionnaire 2, NICU = neonatal intensive care unit.

of exclusive breastfeeding between control and intervention group (37.6 vs.37.8 weeks, p = 0.056). Rates of partial breastfeeding decreased, and rates of exclusive breastmilk fed infants increased, but the rate of no breastfeeding did not decrease significantly (Table 2).

Of the six clinical practices included in the intervention, maternal self-reported use of three of them were significantly improved in the intervention group: More infants continued daily skin-to-skin contact after incubator care (83.2% vs. 88.3%, p = 0.035), less infants used a nipple shield (61.8% vs. 54.2%, p = 0.029), and more mothers initiated breast milk expression before six hours post-partum (32.6% vs. 42.4%, p = 0.007, Table 3). There were no significant

**Table 1. Characteristics of participating infants and mothers.**

|  | Control group | | Intervention group | | |
|---|---|---|---|---|---|
|  | n/N | % | n/N | % | Pearson Chi-Square |
| **Infants** |  |  |  |  |  |
| Participating infants, N | 421 |  | 494 |  |  |
| Gestational age, <28 weeks | 39/421 | 9.3 | 27/494 | 5.5 | 0.027 |
| 28–31 weeks | 103/421 | 24.5 | 98/494 | 19.8 | 0.092 |
| 32–34 weeks | 169/421 | 40.1 | 227/494 | 46.0 | 0.077 |
| 35–36 weeks | 110/421 | 26.1 | 142/494 | 28.7 | 0.377 |
| Multiples | 106/421 | 25.2 | 143/494 | 28.9 | 0,202 |
| Boys | 237/421 | 56.3 | 300/494 | 60.7 | 0.175 |
| **Mothers** |  |  |  |  |  |
| Participating mothers, N | 370 |  | 423 |  |  |
| C-section | 198/362 | 54,7 | 211/421 | 50.1 | 0.201 |
| Planned to breastfeed (dicotom) | 347/360 | 96,4 | 409/420 | 97.4 | 0.424 |
| Breastfeeding experience >4 months exclusive | 63/359 | 17.5 | 76/420 | 18.1 | 0.843 |
| Firsttime mothers | 223/359 | 62.1 | 277/420 | 66.0 | 0.266 |
| Age, >30 years | 171/360 | 47.5 | 186/420 | 44.3 | 0.369 |
| Body mass index, >25 | 128/359 | 35.7 | 149/419 | 35.6 | 0.978 |
| Speaks a Scandinavian language at home | 348/360 | 96.7 | 404/419 | 96.4 | 0.851 |
| Education, low | 81/360 | 22.5 | 90/420 | 21.4 | 0.718 |
| Intermediate | 179/360 | 49.7 | 216/420 | 51.4 | 0.635 |
| High | 100/360 | 27.8 | 114/420 | 27.1 | 0.843 |
| Smoking | 19/360 | 5.3 | 20/420 | 4.8 | 0.742 |

**Table 2. Method and type of infant feeding at discharge.**

| | Control group | Intervention group | Pearson Chi-Square |
|---|---|---|---|
| **Method of feeding** | | | |
| Exclusive breastfeeding, n/N (%) | 244/421 (58.0) | 329/494 (66.6) | 0.007 |
| Partial breastfeeding, n/N (%) | 54/421 (12.8) | 38/494 (7.7) | 0.010 |
| No breastfeeding, n/N (%) | 123/421 (29.2) | 127/494 (25.7) | 0.235 |
| **Type of feeding** | | | |
| Exclusive breastmilk fed, n/N (%) | 283/421 (67.2) | 362/494 (73.3) | 0.045 |
| Partial breastmilk fed, n/N (%) | 67/421 (15.9) | 51/494 (10.3) | 0.012 |
| Not breastmilk fed, n/N (%) | 71/421 (16.9) | 81/494 (16.4) | 0.850 |
| **Timing of feeding** | | | |
| Postmenstrual age at establishment of exclusive breastfeeding, mean weeks (95% CI)* | 37.57 (37.38; 37.76) | 37.81 (37.65; 37.99) | 0.056 |

*Postmenstrual age was available for 236/244 infants in the control group and 316/329 infants in the intervention group.

differences in pacifier use, use of test-weighing, and mothers rooming-in between control group and intervention group.

The results of the sensitivity analyses did not differ from the main results (see S1 Table with Tables 2B and 3B).

Exclusive breastfeeding rates improved in all GA groups, but the study was not powered to detect significant differences in subgroups (Table 4).

A total of 462 (72%) neonatal nurses from the 12 NICUs participated in the one-day training program, that was between 37% and 100% of neonatal nurses from each NICU (Table 5). Changes of breastfeeding rates in the NICUs differed from -8.8 percentage points to 29.6 percentage points. There was a significant correlation between percentage of neonatal nurses participating in the breastfeeding training program and changes in exclusive breastfeeding rates, Pearson Correlation 0.638, $R^2$ Linear 0.361, p = 0.047 (Fig 2).

Breastfeeding meetings for neonatal nurses were implemented in nine out of 12 of the NICUs, one of the nine NICUs did not register duration, participants and topics. Because some NICUs had many breastfeeding meetings each week, but with 1–3 participants, and other NICUs had weekly meetings with up to 15 participants, numbers are given per neonatal

**Table 3. Maternal self-reported practices in control and intervention groups.**

| | Control group n/N (%) | Intervention group n/N (%) | Pearson Chi-Square | Topic discussed in % of "BF-meetings"*** |
|---|---|---|---|---|
| **Practice related to the infant*** | | | | |
| Used a nipple shield | 240/390 (61.5) | 248/460 (53.9) | 0.025 | 38 |
| Minimized use of pacifier during breastfeeding establishment | 218/393 (55.5) | 263/483 (54.5) | 0.763 | 31 |
| Daily skin-to-skin contact after incubator care | 319/383 (83.3) | 414/469 (88.3) | 0.037 | 46 |
| **Practice related to the mother**** | | | | |
| First breastmilk expression before 6 hours of delivery | 111/338 (32.8) | 165/389 (42.4) | 0.008 | 27 |
| Rooming-in for the whole NICU stay | 157/349 (45.0) | 164/410 (40.0) | 0.166 | 16 |
| Used test-weighing at most breastfeeds | 121/330 (36.7) | 119/387 (30.7) | 0.094 | 14 |

* Calculated from a maximum of 421 and 494 infants, respectively in the control and intervention group.

** Calculated from a maximum of 370 and 423 mothers, respectively in the control and intervention group.

*** Mean of wards.

**Table 4. Exclusive breastfeeding rates according to gestational age groups.**

| Gestational age groups | Excl. breastfeeding control group n/N (%)[*] | Excl. breastfeeding intervention group n/N (%)[*] |
|---|---|---|
| <28 weeks | 14/39 (35.9) | 10/27 (37.0) |
| 28–31 weeks | 47/103 (45.6) | 55/98 (56.1) |
| 32–34 weeks | 110/169 (65.1) | 161/227 (70.9) |
| 35–36 weeks | 73/110 (66.4) | 103/142 (72.5) |

[*] Linear-by-Linear association p<0.0001.

nurses in the NICUs. The attendance nurse ratio was between 2.1 and 7.3 breastfeeding meetings per nurse in each NICU, and the duration varied between 10 and 42 minutes per neonatal nurse in each NICU (Table 5). All the six practices were discussed with different frequency at breastfeeding meetings (Table 4). The practices that were less discussed by nurses were among the maternal self-reported practices that did not improve, and the most discussed practices were among the practices that did improve significantly. There was a linear but not significant correlation between topics discussed in breastfeeding meetings and improvement of the clinical practice (Pearson Correlation 0.736, $R^2$ Linear 0.541, p = 0.096, Fig 3).

## Discussion

We found that exclusive breastfeeding rates in preterm infants at discharge improved after training neonatal nurses in six breastfeeding-supportive clinical practices, whereas no difference was found in preterm infants' postmenstrual age at establishment of exclusive breastfeeding. Maternal self-reported use of nipple shield, daily skin-to-skin contact, and first breast milk expression < 6 hours after delivery also improved. Furthermore, we found a significant correlation between the proportion of neonatal nurses participating in the training program and the improvement of exclusive breastfeeding rates in the NICU.

### Exclusive breastfeeding improvements

Our findings indicate that improvements in exclusive breastfeeding outcomes can be achieved by training in evidence-based practice. Similarly, the training programs in the Baby-friendly Hospital Initiative (BFHI) include several of the six clinical practices; rooming-in, early breastfeeding (lactation), skin-to-skin contact, and pacifier use, and have earlier been shown to improve breastfeeding rates [36]. A systematic review of 58 studies and reports found that compliance with the BFHI Ten Steps had a positive impact on breastfeeding outcomes. They also found a dose-response relationship between the number of implemented BFHI steps and early breastfeeding initiation and exclusive breastfeeding at hospital discharge [36]. Also, breastfeeding (and breastmilk feeding) rates in NICUs has formerly been found to increase by implementing multicomponent breastfeeding programs in NICUs or by implementing BFHI in the maternity wards [37–39]. The improved breastfeeding rates in our study are additionally supported by a meta-analysis of 27 randomized controlled trials which found positive effect of breastfeeding promoting interventions on exclusive breastfeeding rates up to six months age, especially the multicomponent BFHI, other combined interventions, and interventions involving healthcare professionals [40]. However, in the present study it cannot be verified if or which of the three improved maternal self-reported practices: early breastmilk expression, continued skin-to-skin contact, or less use of nipple shield contributed most to increase the exclusive breastfeeding rate at discharge. An association between improved practices and improved breastfeeding has previously been found in other studies: A quality improvement

**Table 5. Breastfeeding rates, training, and breastfeeding meeting in NICUs.**

| NICU | Level of care* | N infants, CG | Excl BF at discharge, CG (%) | N infants, IG | Excl BF at discharge, IG (%) | Change from CG to IG | Neonatal nurses participated in training, n/N (%) | Implemented BF meetings | BF meetings, N | BF meetings total participants | BF meetings total duration | BF meeting attendance nurse ratio** | BF meeting duration nurse ratio*** |
|---|---|---|---|---|---|---|---|---|---|---|---|---|---|
| 1 | 3B | 70 | 47,1 | 88 | 70,5 | 23,4 | 67/71 (94.4) | Yes | 112 | 286 | 2950 | 4.0 | 42 |
| 2 | 3B | 46 | 63 | 56 | 58,9 | -4,1 | 50/110 (45.5) | Yes | 87 | 231 | 1357 | 2.1 | 12 |
| 3 | 3A | 46 | 60,9 | 45 | 73,3 | 12,4 | 16/23 (69.7) | Yes | 22 | 88 | 352 | 3.8 | 15 |
| 4 | 3A | - | - | 33 | 69,7 | - | 27/27 (100) | Yes | 24 | 129 | 306 | 4.8 | 11 |
| 5 | 3A | - | - | 41 | 51,2 | - | 28/33 (84.8) | Yes | 19 | 126 | 330 | 3.8 | 10 |
| 6 | 3A | 25 | 40 | 14 | 64,3 | 24,3 | 33/40 (82.5) | No | | | | - | - |
| 7 | 3A | 29 | 62,1 | 31 | 67,7 | 5,6 | 28/38 (73.7) | Yes | 37 | 242 | 934 | 6.4 | 25 |
| 8 | 3B | 44 | 72,7 | 36 | 63,9 | -8,8 | 29/78 (37.2) | No | | | | - | - |
| 9 | 1 | 8 | 37,5 | 10 | 60,0 | 22,5 | 9/23 (39.1) | No | | | | - | - |
| 10 | 3A | 18 | 55,6 | 27 | 85,2 | 29,6 | 26/29 (90.0) | Yes | 35 | 212 | 859 | 7.3 | 30 |
| 11 | 3A | 47 | 59,6 | 67 | 64,2 | 4,6 | 30/45 (66.7) | Yes | | | | - | - |
| 12 | 3A | 23 | 60,9 | - | - | - | - | - | | | | - | - |
| 13 | 3C | 37 | 56,8 | 46 | 69,6 | 12,8 | 119/123 (96.7) | Yes | 64 | 614 | 1854 | 5.0 | 15 |

Excl = exclusive, BF = breastfeeding, CG = control group, IG = intervention group.

*Definitions of level of neonatal care.

Level 1 = Basic care of stable infants born at 35 to less than 37 weeks gestation.

Level 2 = Specialty care of infants born at least 32 weeks gestation or 1,500 grams, with possibility of brief mechanical ventilation or CPAP.

Level 3A = Subspecialty intensive care of infants born at least 28 weeks gestation or 1,000 grams with possibility of mechanical ventilation.

Level 3B = Subspecialty intensive care of infants born at less than 28 weeks gestation or 1,000 grams, with possibility of advanced respiratory support, and access to pediatric surgical specialist.

Level 3C = As level 3B but including extracorporeal membrane oxygenation and surgical repair of complex congenital cardiac malformations.

**Total participants in BF meetings/ number of nurses in ward.

***Total duration of BF meetings/ number of nurses in ward.

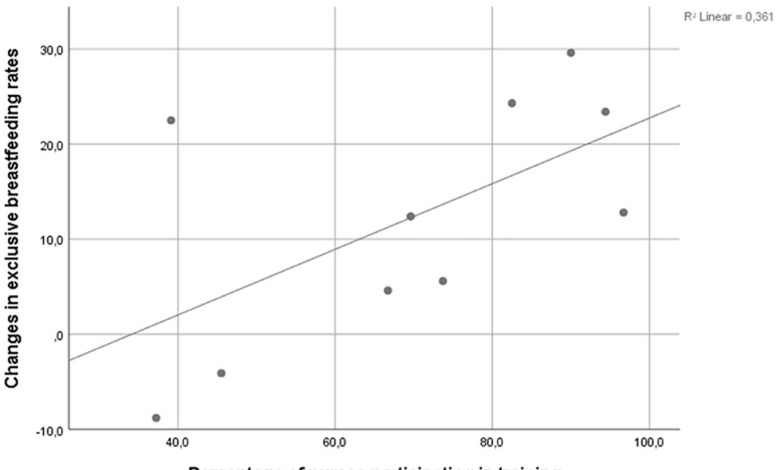

**Fig 2. Changes in exclusive breastfeeding rates in NICUs according to percentage of nurses participating in training.**

study from the U.S. showed that improved counselling of mothers of very preterm infants decreased the time to first breast milk expression and increased the proportion of infants receiving exclusively maternal breast milk at 28 days of life. The study did not, however, investigate the association with exclusive breastfeeding at discharge [17]. Furthermore, a Brazilian pre-and post-intervention study showed significant increase in exclusive breastmilk feeding rates after implementation of Kangaroo mother care/skin-to-skin contact in a NICU, but they did not investigate exclusive direct breastfeeding [41]. To our knowledge, no studies have been performed on how to reduce the use of nipple shields in preterm infants, nor whether a reduction in use would influence exclusive breastfeeding at discharge. Our study suggests an association between training nurses in breastfeeding-supportive practices, improved maternal self-reported use of the practices in the NICU, and improved exclusive breastfeeding in preterm infants.

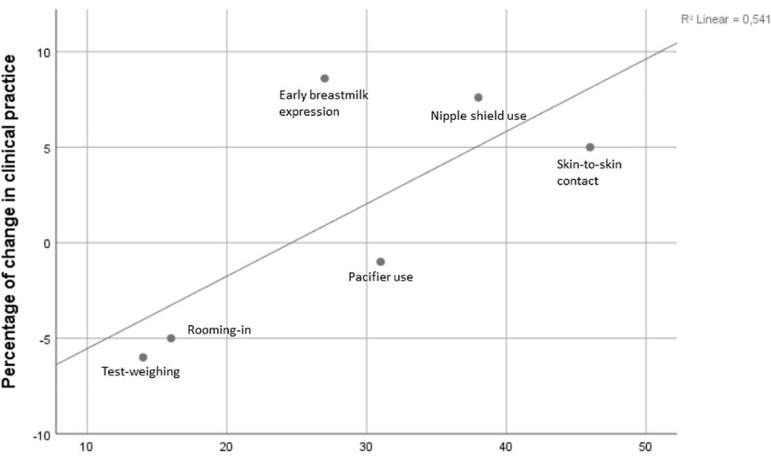

**Fig 3. Percentage of change in clinical practice by practice discussed in percentage of breastfeeding meetings.**

Rooming-in and minimizing use of pacifier, which have previously been associated with earlier establishment of exclusive breastfeeding [14], did not improve after the training. Even though all NICUs had some available beds for mothers, most did not increase the number of mothers staying overnight in the same room as their infant during the whole NICU stay. Continued daily skin-to-skin contact increased from 83% in the control group to 88% in the intervention group. As the rate of daily skin-to-skin contact was already high, this increase may not have had much impact on postmenstrual age at establishment of exclusive breastfeeding. We were not able to find a decrease in postmenstrual age at establishment of exclusive breastfeeding. This may partially be due to ceiling effect of skin-to-skin contact and may partially be influenced by the failure to improve minimization of pacifier use and the rate of rooming-in, and partly by the limitations of a quasi-experimental design.

## Training of neonatal nurses

The training program in the present study consisted of 7 hours which was less than other intervention studies that have shown to improve breastfeeding rates: Nilsson et al. used 11 hours in their training program [42] and Lowson et al. 7 days [31]. We believe that actively involving the neonatal nurses in the training sessions and integrating their ideas from the brainstorms on how to improve the six clinical practices in their NICU strengthened their active participation and motivation to optimize guidance in the practices and was important to the increase in breastfeeding rates. Change in practice must grow from the existing environment to succeed, and staff reflection as part of an education program has previously shown sustainable success when implementing skin-to-skin contact [43]. A UK multicentre study in 18 neonatal units also used an approach tailored to the needs of the staff in the individual neonatal unit and succeeded in increasing skin-to-skin sessions and breast milk feeding [31]. The correlation between neonatal nurse participation in the breastfeeding training program and improvements in breastfeeding in NICUs points out the importance of including all health care staff in the training when implementing evidence-based practice. We found a tendency but not significant correlation between how frequently the six practices were discussed at the breastfeeding meetings and the improvement of the six practices. The lack of significance could be due to the small number of practices. A clinical effect could still be possible. We believe the reflection on own practice during breastfeeding meetings had an effect in adjusting and maintaining optimal practice.

Surprisingly, the exclusive breastfeeding rate at discharge in the control group was only 58%. Although we had not expected a decrease in exclusive breastfeeding in the Danish preterm population from 68% reported in 2009–2011, an even more profound decrease from 59% to 45% has been found in Sweden in the period 2004 to 2013 [7]. This reinforces that improving breastfeeding rates needs constant focus.

## Strengths and limitations

The strengths of the present study are the high number of participants, the multicentre design, and the adherence to the pre-post-design: No mother-infant dyads in the control group were exposed to the intervention and all mother-infant dyads in the intervention group were from NICUs participating in the intervention phase. Another strength was the narrow definition of breastfeeding as exclusive and direct breastfeeding. Clinical practices such as nipple shield use and pacifier use could have an impact on the infant's ability to establish exclusive breastfeeding whereas the mother's milk supply might not be affected. As a result, exclusive breastfeeding was the primary outcome of interest rather than exclusive breastmilk feeding or any breastfeeding. The clinical practices were not measured according to whether nurses delivered

information to mothers, but rather according to whether mothers adhered to the recommended practices. The intervention was not regarded as implemented until it resulted in optimal practice reported by mothers. Thus, compliance with the six clinical practices was only measured by mothers' questionnaire responses.

In a quasi-experimental design, we cannot eliminate that other changes could cause the difference. However, the rather short distance between pre-and post-intervention phases increases the likelihood of the intervention to cause the increase in breastfeeding rates. Randomizing mothers-infant dyads to control or intervention would have been preferable but not possible, as a spillover effect could not be avoided. Improvement in breastfeeding rates between control and intervention groups could have been due to fewer extremely preterm infants in the intervention group who are known to be breastfed less [8] as we also found. However, subgroup analyses showed that exclusive breastfeeding rates increased in all four GA groups, why the improvement between control and intervention groups was not only explained by fewer extremely preterm infants. Not all NICUs participated in both pre- and post-interventions and differences in breastfeeding rates between NICUs could bias the results. Nevertheless, in the 2009–2011 cohort, the breastfeeding rate in Danish NICUs participating in the later pre-intervention was not significantly different from those not participating in the pre-intervention (66.4 vs. 69.4, p = 0.218), and the same applied for NICUs participating/not participating in the post-intervention (66.9 vs. 69.0, p = 0.405). We showed in this study that in a context with relatively high breastfeeding rates, exclusive breastfeeding rates were improved in the intervention group, while the rates of no breastfeeding were not improved. It is not known whether the present intervention would only improve exclusive breastfeeding or also no breastfeeding in contexts with lower breastfeeding rates.

Even though efforts were made to increase response rate, several participants did not complete the study. However, the response rates were similar between control and intervention groups (74% and 73%, respectively). Some participants who answered both questionnaires did not answer all questions and consequently numbers of participants included in each calculation differs. The question about test-weighing had the most missing values (10.8% missing, primarily from the second twin) but we have no reason to believe that the missing answers were systematic and would have affected the results substantially.

## Conclusion

Exclusive breastfeeding rates in preterm infants could be improved by training neonatal nurses in evidence-based breastfeeding support. It is possible to transform evidence-based knowledge into practice and increase rates of early breastmilk expression, continued skin-to-skin contact, and reduce the use of nipple shields. A significant correlation between the proportion of neonatal nurses participating in the training program and the improvement of exclusive breastfeeding rates in the NICU calls for the importance of including all health care staff when implementing evidence-based practice to ensure positive effect on exclusive breastfeeding rates. The results could be beneficial for preterm mother-infant dyads and increase the quality in clinical nursing practice to these fragile groups of mothers and infants.

## Supporting information

**S1 File. Breastfeeding support training program for neonatal nurses.**
(PDF)

**S2 File. Poster for parents with short statements.**
(PDF)

**S3 File. Questionnaire 1.** The nutrition study of preterm infants 2016–2019.
(PDF)

**S4 File. Questionnaire 2.** The nutrition study of preterm infants 2016–2019.
(PDF)

**S1 Table. Sensitivity analyses.** Sensitivity analyses of primary and secondary outcomes.
(PDF)

## Acknowledgments

We want to thank all the participating mothers for contributing data to the study, the National Expert Panel in Breastfeeding Infants with Special Needs for contributing to the development of the study, and the nurse managers and contact persons in the participating Danish NICUs for supporting the study and enrolling participants.

## Author Contributions

**Conceptualization:** Ragnhild Maastrup, Helle B. Sandfeld, Hanne Kronborg.

**Data curation:** Ragnhild Maastrup, Sisse Walloee.

**Formal analysis:** Ragnhild Maastrup, Ane L. Rom.

**Funding acquisition:** Ragnhild Maastrup.

**Project administration:** Ragnhild Maastrup.

**Supervision:** Hanne Kronborg.

**Writing – original draft:** Ragnhild Maastrup, Ane L. Rom.

**Writing – review & editing:** Ragnhild Maastrup, Ane L. Rom, Sisse Walloee, Helle B. Sandfeld, Hanne Kronborg.

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
