## [Decision Letter · Decision Letter 0]

9 Oct 2020

PONE-D-20-24962

Improved exclusive breastfeeding rates in preterm infants after a neonatal nurse training program focusing on six breastfeeding-supportive clinical practices.

PLOS ONE

Dear Dr. Maastrup,

Thank you for submitting your manuscript to PLOS ONE. After careful consideration, we feel that it has merit but does not fully meet PLOS ONE’s publication criteria as it currently stands. Therefore, we invite you to submit a revised version of the manuscript that addresses the points raised during the review process. The reviewer #1 raised several major concerns that should be addressed in a revision of your manuscript.

We look forward to receiving your revised manuscript.

Kind regards,

Olivier Baud, MD, PhD

Academic Editor

PLOS ONE

Journal Requirements:

2. Please attach a Supplemental file of the training materials used in this study or a link if it's published online.

Reviewers' comments:

Reviewer's Responses to Questions

**Comments to the Author**

1. Is the manuscript technically sound, and do the data support the conclusions?

Reviewer #1: Yes

Reviewer #2: Yes

2. Has the statistical analysis been performed appropriately and rigorously? 

Reviewer #1: I Don't Know

Reviewer #2: Yes

3. Have the authors made all data underlying the findings in their manuscript fully available?

Reviewer #1: Yes

Reviewer #2: Yes

4. Is the manuscript presented in an intelligible fashion and written in standard English?

Reviewer #1: Yes

Reviewer #2: Yes

5. Review Comments to the Author

Reviewer #1: In this paper, Maastrup et al. report a quasi-experimental study conducted in 13 Danish NICUs and aiming to assess a training program for NICU nurses, focusing on 6 breastfeeding (BF) supporting practices. In all, 420 and 294 preterm mother-infant dyads were enrolled in the before and after intervention periods, respectively. Results show a significant increase in BF rates of preterm infants at discharge in the post-intervention phase, associated to a mothers self-reported increase in 3 of the practices (skin to skin, reduction in nipple shield use, and early 1st expression of milk after birth). No difference was found in postmenstrual age at establishment of exclusive BF. Interestingly, there was a documented correlation between rates of trained nurses and changes in BF rates.

This well written paper is very relevant regarding the crucial importance to increase BF rates in this vulnerable population, and the existing need for evidence-based, efficient and feasible interventions, such as the proposed in this elegant study. It also has the advantage to document adherence to the intervention in the participating centres, reflecting its feasibility. Thus, these valuable results worth to be shared and published.

Nevertheless, several points deserve further information. I have the following questions, comments and concerns, to address to the authors:

1. The study rational, methods, results and discussion focuses only on exclusive BF rates. It could be interesting to bring some considerations on any BF rates, as even partial BF is still beneficial: what were the rates of total (exclusive and partial) BF? Did the authors record them? If so, did they observe any difference after the intervention?

2. Also, the main outcome “exclusive BF” was defined in this study as “infant feeding directly at and from the breast” (page 5, line 95), which differs from the WHO definition (“no other food or drink, not even water, except breast milk, including milk expressed or from a wet nurse”, see also https://www.who.int/nutrition/topics/infantfeeding_recommendation/en/). I do not understand the choice of this rather restrictive definition. In all cases, it would be important to provide additional information about potential differences in reported rates of exclusive BF when using the WHO definition, which remains a reference in reporting and comparing BF rates.

3. Why the authors choose to assess BF rates at discharge? Information on BF duration or at least on BF continuation after a couple of weeks/months at home would also have been an added value, regarding the dose effects on one hand and the known risk of BF discontinuation at discharge on the other.

4. What were the reasons for restricting the training to nurses rather incorporating all the health caregivers, including physicians? There is indeed a global and recognized need to improve medical education and involvement in BF support, a growing interest to conduct interdisciplinary actions, especially for BF issues, and to avoid discordant information to the mothers. Such a training program could have been a synergic opportunity to reach these goals too.

5. Although multicentric, the study is not international, but limited to the Denmark. As in other Scandinavian settings, BF epidemiology and determinant factors in DK are not easy to generalize to the rest of the Europe and other countries. Thus, some information on BF policy in DK, such as duration of the maternity leave or BF rates in the general population, could help in transposition of the results and comparisons with other settings.

6. Study design: in order to avoid cross-contamination, the study followed a quasi-experiment design. However, a some more robust alternative in this situation could have been to use a stepped wedge cluster randomised trial (Hemmin, BMJ 2015;350:h391/ doi:https://doi.org/10.1136/bmj.h391). Dis the investigators consider this possibility in planning the study, and, if so, why was this strategy not retained?

7. Outcomes: as the study sample was calculated on the basis on the exclusive BF rates increase, this should, on my opinion, be the only primary outcome. Others are secondary outcomes.

8. Methods, pages 7-8: The description of the interventional training program for neonatal nurses reports here 5 of the 6 clinical practices, but not the rooming in. Why? Conversely, it comprises the extra attention to give to groups at particular risk of not breastfeeding exclusively, which could also be consider as an additional action on practices.

9. Although interesting and increasing, early discharge policies with at home care and follow up are not available in most countries. Was the duration of total care (hospital stay and home follow up) recorded and compared between the two periods? This would be interesting from an economical point of view also, as shorter durations would be associated to lower costs, which can make a sound argument for health policies establishments.

10. More description on participating centres would be useful: except for the four level III, what were the other NICUs levels? How many yearly admissions (min-max)? Among them, were they any Baby Friendly Initiative Hospitals? Availability and use of donor human milk?

11. Regarding the rooming-in, parental facilities available accommodations policies in centres would need to be reported too. What kind of accommodations/rooms were offered? Were they similar in all participating centres?

12. Was maternal satisfaction assessed and compared between the 2 periods? It would be an important outcome too.

13. Results, table 5: There were huge differences in nurses participating rates, from one third to 100%. Moreover, 2 centers could not realize the intervention at all. What were the reasons for such differences? What were the putative difficulties in centres with lower adherence? This would help readers to understand and to transpose the feasibility of the intervention.

14. From a methodological and statistical point of view, couldn’t one alternatively consider to include the 2 centres that were unable to perform the intervention in an intention to treat analysis?

15. Did the investigators perform a separate analysis for each center level?

16. A limitation of the study is the relatively high number of lost data, especially in self-reporting questionnaires, encompassing risks of bias, especially reporting bias. Was any action attempted to improve the responding rates (such as phone call by the investigators, others)?

17. The fact that the use of pacifier, in particular, was not impacted is interesting, as it is thus unlikely to have played a determinant role in this study. Authors could briefly discuss this, considering the existing controversies on the use of pacifiers in NICUs, which could also have beneficial effects in neonatal wards, such as reducing pain and stress, favouring non-nutritive suction, or reducing the risk of sudden infant death.

Minor:

18. Abstract, page 2, lines 26-30: what is primary and secondary outcomes should be made explicit and exposed in this order (see also point 7).

19. Negative results, such as the absence of difference in postmenstrual age at establishment of exclusive BF, should also be mentioned.

20. Introduction, page 4, lines 50-51: Other important protective effects of mother’s milk such as BPD or neurodevelopmental advantage could also be mentioned.

21. Methods, page 9, line 172: Three questionnaires are cited, but only two are described and provided in the supplementary material.

22. The whole intervention could also be considered as a “ bundle of care” which are gaining increasing interest, not only in nosocomial infection control, but also in BF and other practices assessment. This could be mentioned somewhere.

23. Training and meetings: did study nurses or coordinators supervise them?

24. Results, Table 1: quantitative variable are presented in categorical ranges, which are not always very intuitive. An additional description (even in text) of a few quantitative variables in terms of means and/or median and min-max range and/or DS would be helpful, at least for Gestational age and Birthweight.

25. Results, flow chart and figures: not sharp in the pdf version.

26. Results, Table 2, 3 and 5: not fully visible in the pdf (right columns are cut).

Reviewer #2: The study was exceptionally developed and implemented, the write up is replicable, and the analysis is thorough. The authors addressed each variable with statistical analysis and eloquent discussion.

My only suggestion is to define exclusive breastfeeding at discharge. Did this mean the infant was exclusively breastfed the same day as discharge, within 24 hours of discharge, or some other measure?

6. PLOS authors have the option to publish the peer review history of their article (what does this mean?). If published, this will include your full peer review and any attached files.

Reviewer #1: No

Reviewer #2: **Yes: **Angela Lober, PhD, RNC, IBCLC

---

## [Author Response · Author response to Decision Letter 0]

20 Nov 2020

We have improved the manuscript according to the reviewer’s comments, which we have answered one by one in the uploaded “Response to reviewers”.

---

## [Decision Letter · Decision Letter 1]

28 Dec 2020

Improved exclusive breastfeeding rates in preterm infants after a neonatal nurse training program focusing on six breastfeeding-supportive clinical practices.

PONE-D-20-24962R1

Dear Dr. Maastrup,

We’re pleased to inform you that your manuscript has been judged scientifically suitable for publication and will be formally accepted for publication once it meets all outstanding technical requirements and the 2 minor revisions raised by the reviewer #1 are adressed.

Kind regards,

Olivier Baud, MD, PhD

Academic Editor

PLOS ONE

Additional Editor Comments (optional):

Reviewers' comments:

Reviewer's Responses to Questions

**Comments to the Author**

1. If the authors have adequately addressed your comments raised in a previous round of review and you feel that this manuscript is now acceptable for publication, you may indicate that here to bypass the “Comments to the Author” section, enter your conflict of interest statement in the “Confidential to Editor” section, and submit your "Accept" recommendation.

Reviewer #1: All comments have been addressed

Reviewer #2: All comments have been addressed

2. Is the manuscript technically sound, and do the data support the conclusions?

Reviewer #1: Yes

Reviewer #2: Yes

3. Has the statistical analysis been performed appropriately and rigorously? 

Reviewer #1: I Don't Know

Reviewer #2: Yes

4. Have the authors made all data underlying the findings in their manuscript fully available?

Reviewer #1: Yes

Reviewer #2: Yes

5. Is the manuscript presented in an intelligible fashion and written in standard English?

Reviewer #1: Yes

Reviewer #2: Yes

6. Review Comments to the Author

Reviewer #1: This manuscript is relevant, interesting and well written, and desserves publication.

Authors answered in detail to all adressed comments, and integrated most of them in the revised version.

However, on my opinion, they are still 2 points that should beTconsidered in the manuscript:

1. Table 2 shows that the intervention improves Exclusive BF as e method of feeding, but not any breastfeeding, neither any breastmilk fed rates. This is important as the rates of BF are admirably high in this study,a while they are lower in many other countries/NICUs, that probably need to implement measures that increase overall BF rates bore to be able to reproduce such interventions and results.

2. The 2 NICUs that were intended to participate but unable to implement the intervention were considered in the control group. So this is not an "intention to treat" analysis, and does not reflect the "real life" feasability of the intervention. It would be important to present aslo an intention to treat analysis at least on the primary outcome to adress this point.

Reviewer #2: Excellent revision. All my questions regarding methodology and results were addressed. I have no further questions or comments.

7. PLOS authors have the option to publish the peer review history of their article (what does this mean?). If published, this will include your full peer review and any attached files.

Reviewer #1: No

Reviewer #2: No

---

## [Editor Report · Acceptance letter]

20 Jan 2021

PONE-D-20-24962R1 

Improved exclusive breastfeeding rates in preterm infants after a neonatal nurse training program focusing on six breastfeeding-supportive clinical practices. 

Dear Dr. Maastrup:

I'm pleased to inform you that your manuscript has been deemed suitable for publication in PLOS ONE. Congratulations! Your manuscript is now with our production department. 

Kind regards, 

on behalf of

Pr. Olivier Baud 

Academic Editor

PLOS ONE